# Assessment of COVID-19 Vaccine Acceptance and Its Associated Factors during the Crisis: A Community-Based Cross-Sectional Study in Benin

**DOI:** 10.3390/vaccines11061104

**Published:** 2023-06-16

**Authors:** Sètondji Géraud Roméo Padonou, Clément Kakaï Glèlè, Manfred Accrombessi, Bayode Romeo Adegbite, Edouard Dangbenon, Houssaïnatou Bah, Enangnon Akogbeto, Ali Imorou Bah Chabi, Landry Kaucley, Salifou Sourakatou, Ange Dossou, Achille Batonon, Tania Bissouma-Ledjou, Benjamin Hounkpatin

**Affiliations:** 1Department of Public Health, Faculty of Health Sciences, University of Abomey-Calavi, Cotonou 01 BP 526, Benin; 2Country Office, World Health Organization, Cotonou 01BP 918, Benin; 3Disease Control Department, London School of Hygiene and Tropical Medicine, Faculty of Infectious and Tropical Diseases, London WC1E 7HT, UK; 4African Public Health Consulting and Research Group, Cotonou, Benin; 5Cotonou Entomological Research Center, Ministry of Health, Cotonou 06 BP 2604, Benin; 6Alliance for the Promotion of Community Health, Research and Scientific Innovation, Cotonou, Benin; 7Center of Tropical Medicine and Travel Medicine, Department of Infectious Diseases, Amsterdam Public Health, Amsterdam Infection and Immunity, University of Amsterdam, Amsterdam University Medical Centers, Location AMC, Meibergdreef 9, 1105 AZ Amsterdam, The Netherlands; 8Centre de Recherches Médicales de Lambaréné, Lambarene BP242, Gabon; 9Ministry of Health, Cotonou 01 BP 882, Benin; 10Faculté des Sciences de la Santé, Université d’Abomey-Calavi, Cotonou 01 BP 526, Benin

**Keywords:** COVID-19 vaccine acceptance, associated factors, sub-Saharan Africa

## Abstract

Background: Having a maximum number of people vaccinated was the objective to control the COVID-19 pandemic. We report in this manuscript the factors associated with the willingness to be vaccinated against COVID-19 during the pandemic period. Methods: From April to May 2022, a community-based cross-sectional survey was performed. Participants were randomly selected from four districts in Benin (taking into account the COVID-19 prevalence). Mixed-effect logistic regression models were used to identify the variables associated with COVID-19 vaccine acceptance. Results: A total of 2069 participants were included. The proportion of vaccine acceptance was 43.3%. A total of 24.2% were vaccinated and showed proof of vaccination. The population’s request for vaccination was higher after the third epidemic wave. The district of residence, the education level, a fear of being infected, the channel of information, poor medical conditions, a good knowledge of the transmission mode and symptoms, and good behaviors were significantly associated with vaccine acceptance. Conclusion: The overall acceptance of the COVID-19 vaccine in the Beninese population was relatively high. However, vaccine campaigns in areas with a low acceptance as well as the disclosure of information, particularly on our knowledge of the disease and the safety, side effects, and effectiveness of the COVID-19 vaccines, should be strengthened with adapted and consistent messages.

## 1. Introduction

The COVID-19 pandemic has caused a significant number of deaths worldwide and nobody was warned about or prepared for such a pandemic. The World Health Organization (WHO) has globally identified 640,395,651 confirmed cases of COVID-19 and 6,618,579 deaths (2–3% case fatality rate) as of December 2022 [1]. African countries have also been affected by the COVID-19 pandemic, with a total of 9,402,777 cumulative cases and 1,071,245 cumulative deaths [1]. This pandemic highly affected the worldwide health system with socio-economic consequences [2]. An important challenge for public health decision makers in low-income countries was to define adapted socio-economic measures to prevent the spread of the virus. Vaccination is an effective tool to reduce the burden of COVID-19, but its success depends on vaccine coverage and effectiveness [3]. There was much skepticism about vaccination against COVID-19, even among some public health stakeholders and the scientific community. As of June 2023, 13.42 billion COVID-19 doses have been administered globally, and 30.2% of people in low-income countries have received at least one dose [4]. Vaccine acceptance factors include comprehensive and consistent messages from public health authorities, compliance, the establishment of a confidence environment, effective vaccination campaigns, and sustained health system capabilities [5,6,7,8]. Addressing the vaccine hesitancy concerns is essential to avoid vaccine program failure, particularly in rural communities. Therefore, the Ministry of Public Health and other stakeholders should implement an adapted framework to better understand the doubts and concerns of the population and to develop the best approach to improve COVID-19 immunization rates.

The first case of COVID-19 in Benin was identified in March 2020 [9]. Benin is a West African country with an estimated population size of 12,123,200 inhabitants in 2020 [10]. As of 31 December 2022, according to the National Direction of Public Health, 1,147,915 people were screened, including 27,986 people with confirmed PCR-positive cases, 96,195 people with suspected cases, 27,821 (99.00%) cured people, and 163 deaths [1]. Of all the confirmed cases, 47.3% of those affected were men compared to 52.7% that were women. The age group of 15 to 45 years was the most affected (65.5%) [9]. The average positivity index (number of positive cases/number of cases detected) was 2.4%. The Littoral district (Cotonou, the economic capital city) was the most affected [9]. To contain COVID-19, the Benin government’s prevention strategies included screening all flight passengers coming into the country. The Ministry of Health installed a temperature scanner, a handwashing apparatus, and an isolation room in the international airport. There was no general lockdown. However, a sanitary cordon was established around the city of Porto-Novo where the first case was detected to control the spread of the virus throughout the rest of the country [11]. The overall vaccination coverage was 3% in 2021, and was insufficient to provide immunity to the general population. With the support of the WHO, the government decided to organize an intensive vaccination campaign and initiated a survey to identify factors associated with COVID-19 vaccine acceptance and to plan. Based on the evidence and vaccines doses available during the pandemic period, people aged 12 years old and over were eligible to be vaccinated for free. We report in this study the COVID-19 vaccine acceptance during the crisis (third wave) and the associated factors in the Beninese population.

## 2. Materials and Methods

### 2.1. Study Area and Design

The study was a community-based cross-sectional study conducted in four districts in Benin (Cotonou, Abomey-Calavi, Porto-Novo, and Djougou) from April to May 2022. The districts of Cotonou, Abomey-Calavi, and Porto-Novo, located in southern Benin, were selected because of the large number of COVID-19 cases recorded, with Cotonou’s district being the hotspot of the outbreak in Benin. Djougou’s district, located in the north of Benin, was among the least affected.

### 2.2. Study Population and Sample Size

All household members aged 12 years and older that were permanent residents in the study area and that gave written informed consent were included in the study. Assent was sought for children from 12 to 18 years of age. The estimated sample size was 2000 participants to ensure a good power. This sample size was calculated based on an estimated proportion of acceptance of 50% (the expected result was unknown, but was guesstimated to be about 50% as recommended in such cases), with a confidence level of 95% and a precision of 5%. Since this was a cluster survey, a design effect of 5 was applied. Based on these parameters, the minimum sample size was 1921. This number was increased to at least 2000 participants to take into account any cases of consent withdrawal or refusal to participate. The sample size was calculated using OpenEpi, version 3, an open-source calculator [12].

### 2.3. Participant Selection

We carried out two-stage random sampling in each district. The Benin demographic and health survey’s standardized method suggested by Benin’s National Institute of Statistics and Economic Analysis (INSAE) was used [13]. Briefly, the unit of sampling in each district was a household. The INSAE grouped the households in each district into smaller enumeration areas (EAs). The first step was the random selection of the EAs in each district and the second step was the household’s selection. All households in each selected EA were visited. The number of EAs and households to be selected in each district was proportional to the district population size based on a recent Benin demographic and health survey.

### 2.4. Study Questionnaire and Data Collection Procedures

A structured, standardized, and validated questionnaire was administered face-to-face to one household member randomly selected in each household. The questionnaire used for this survey was derived from a template suggested by the WHO working group for such a survey during the COVID-19 pandemic. This WHO questionnaire was modified slightly and adapted to the study context and objectives. The questionnaire was submitted to the experts appointed by the Benin country’s WHO office for review and validation before implementation. A pilot phase lasted seven days from 19 to 25 April 2022 in the form of a pre-test, which was performed to check if the questionnaire was easily understandable by the participants and interviewers. Each interviewer administered the questionnaire to 10 different households. This phase allowed us to identify the challenges related to the survey (length of the questionnaire or difficulties in taking GPS coordinates), to readjust, and to finalize the questionnaire for data collection. The mean duration of an interview was 25 min. The questionnaire is provided as Appendix A.

The survey was carried out in three phases: the training phase, the pilot phase, and the actual survey phase. A total of 25 field investigators (interviewers) were recruited for this study; they were medical students and people with a bachelor’s or master’s degree in community health, public health, or socio-sanitary sciences. They were trained on the study protocol, the questionnaire, informed consent, and the data collection procedure.

Data collection took place over 10 days. Before starting the administration of the questionnaire in the households, the interviewers visited each village’s local authorities to inform them about the study. The local authorities appointed local guides or community health workers to assist the investigators in identifying the households. Overall, an average of 193 households were visited each day, and each investigator visited an average of 12 households per day.

Data were collected from both the head of the household and the selected member. The data collected were related to the general characteristics of the household and specific COVID-19 questions. The data were based on participant self-reporting and declarations. Only the vaccination status was collected from the vaccination card.

### 2.5. General Data

−Data on the household’s demographic characteristics (including age and sex, education of individuals, and occupation).−Housing characteristics and asset ownership, which were used to build a socioeconomic score.

### 2.6. Specific Data

−Data on the knowledge, attitudes, and practices (KAP) related to COVID-19, including the knowledge of the symptoms and the modes of transmission of COVID-19; the knowledge and use of preventive measures against COVID-19 (e.g., hand washing and physical distancing); and the knowledge of who is at risk for COVID-19 infection.−Data on vaccination against COVID-19 (existing treatment, type of vaccines, and information channel for the vaccine).−Data on the acceptability of being vaccinated against COVID-19 (non-vaccination and unwillingness to be vaccinated, and the reason).

### 2.7. Data Management and Statistical Analysis

All data collected during the survey were recorded in electronic forms on smartphones installed with KoboCollect based on Open Data Kit technology and analyzed with STATA, version 17 (Stata Corp LP, College Station, TX, USA). The main outcome was vaccine acceptance, as defined by participants who were already vaccinated plus those who were not yet vaccinated, but were willing to be vaccinated. Vaccination status was declarative, but proof of vaccination (vaccination card) was asked for by the study field workers to estimate the real vaccine coverage. Descriptive statistics and 95% confidence intervals were used to summarize the demographic data. Mixed-effect logistic regression models were used to assess the factors associated with COVID-19 vaccine acceptance, with clusters included as a random effect. Variables with *p*-values below 0.2 were included in multivariate analyses and were eliminated step-by-step using the backward selection procedure. Only variables with a *p* < 0.05 were retained in the final model. For variables with more than two categories, a *p*-value of the global test is given. The risk factors assessed included characteristics at the household level (residence, number of residents, and household assets), characteristics at the individual level (gender, education level, ethnic group, age, religion, and marital status of the participant), and the channel of information. A household socioeconomic status was determined by using a principal component analysis with the following variables included: type of lighting, access to water, type of roof, type of floor, type of toilet, household head’s level of education, household crowding, and ownership of assets (motorbike, television, bike, radio, sheep, bed, or phone).

## 3. Results

A total of 2318 households were visited; 2245 (96.8%) of the households had at least one adult member present at the time of the visit, 69 (2.9%) households had members that were absent, and 4 (0.2%) households were vacant. Of the 2245 households with members present, 2074 (92.4%) gave their informed consent and 2069 individuals agreed to complete the survey.

### 3.1. Characteristics of the Study Population

Table 1 presents the general characteristics of the study population. The mean number of people living in the households visited was 4.31 (SD: ±2.5, range: 1–25). The mean age of the participants was 37 years (SD: ±14, range: 12–85). Females were more predominant (54%) and most of the participants were educated (83.1%). The predominant COVID-19 channels of information reported were the radio (77.5%), television (69.4%), social networks (46.9%), and word-of-mouth (80.1%).

The most reported poor medical condition was high blood pressure (11%) followed by tobacco smoking (3.8%) (Table 2). Regarding the history of previous COVID-19 infection, very few participants reported being infected (3.7%). On a scale of fear of being infected from 1 to 10, 74.2% of the participants reported a low fear (scale < 6) of COVID-19 infection. Overall, 39.8% of participants had a good knowledge of COVID-19 (Table 2). The proportion of participants with a good knowledge of the mode of transmission, symptoms, attitude, and practices were 32.0%, 46.3%, 26.7%, and 36.4%, respectively.

### 3.2. COVID-19 Vaccine Status and Acceptance Level

Table 3 presents the COVID-19 vaccine status and acceptance. Nearly half of the participants surveyed declared that they were vaccinated (48.7%). However, only 50% of those who reported being vaccinated were able to present their vaccination proof. The proportion of those vaccinated with proof was 24.2%. The most widely administered vaccine was “Johnson & Johnson” (79.8%), followed by “Sinovac” (7.9%), “Pfizer” (7.4%), and “Covishield” (3.4%).

The vaccine acceptance rate was 67.9% based on the number of people declaring to be vaccinated plus those willing to be vaccinated; however, when considering only participants with proof of vaccination, the vaccine acceptance rate decreased to 43.3%. We observed a high heterogeneity between different districts. Cotonou had a lower vaccine acceptance rate (61.3%) compared to Djougou (78.5%), Abomey-Calavi (66.2%), and Porto-Novo (69.4%). The main reasons reported by non-vaccinated participants who did not wish to be vaccinated were a fear of side effects (47.5%) followed by doubts about the effectiveness of the vaccines (44.0%), with a higher proportion in urban areas.

### 3.3. COVID-19 Vaccine Demand Trend

Figure 1 shows the proportion of people who reported being vaccinated over time. We observed that populational vaccine requests increased significantly after the third and longest epidemic wave, between July 2021 and October 2021, but gradually decreased after the fourth wave.

### 3.4. Analysis of Potential Factors

After adjusting for confounding factors (Table 4), vaccine acceptance was significantly associated with the district of residence (with Djougou having the highest probability of acceptance (aOR: 2.70, 95% CI: 1.71–4.28)), a college level of education (aOR: 1.88, 95% CI: 1.25–2.81), a moderate fear of being infected (aOR: 1.60, 95% CI: 1.06–2.41), and receiving information from social networks (aOR: 1.33, 95% CI: 1.05–1.68) or community political leaders (aOR: 1.32, 95% CI: 1.03–1.69) as the channel of information. Poor medical conditions were associated with a high probability of acceptance, particularly high blood pressure (aOR: 1.43, 95% CI: 1.03–1.99), while cardiac disease was related to a lower acceptance (aOR: 0.30, 95% CI: 0.12–0.77). Having a moderate-to-good knowledge of COVID-19 symptoms and the mode of transmission and having good practices were associated with a COVID-19 vaccine acceptance response.

## 4. Discussion

The COVID-19 pandemic has had a devastating public health impact worldwide, motivating the international community (private and government organizations) to work together and find urgent solutions to contain the pandemic. Significant investments never seen before have been made to develop vaccines against COVID-19 [8]. However, an uneven access and skepticism about COVID-19 vaccination have most likely impacted the financial and human resource efforts to control the pandemic [14].

This study, which included 2069 subjects from several cities within the country, aimed to assess the COVID-19 vaccine acceptance rate in Benin. We observed a population-based vaccine acceptance of 67.9% and, after adjustment, 43.4%. This finding was quite similar to those reported for Zimbabwe (55.7%), Ghana (51%), and Nigeria (51.1%) [15,16,17]. However, it was higher than that reported by Mudenda et al. in Zambia (33.4%) [18]. One of the reasons for this difference could be the culture and beliefs of people. Indeed, the hesitation to be vaccinated is also related to the culture and beliefs. A recently published literature review covering 16 African countries reported that the vaccine uptake ranged from 6.9% to 97.9% with considerable heterogeneity depending on the study population (health workers, teachers, students, and the general population) [19]. The lower levels of COVID-19 vaccine acceptance are in contrast to studies conducted in other regions, such as Europe and the Americas [20], Kuwait, and the UK [21,22].

In our study population, the proportion of people who reported being vaccinated was 48.7%. However, about half of those who reported being vaccinated were able to provide proof of vaccination, leading to the vaccination coverage decreasing to 24.2%. Indeed, the vaccination status was confirmed by checking the participant’s vaccination card. The vaccination card was not always available for a variety of reasons (the card was not available at the time of the interview, the card was lost, or the participant refused to show it), likely resulting in an underestimate of the actual vaccine coverage rate. According to statistics from the Ministry of Health and the WHO for Benin, the vaccination coverage was 35.1% during the same period. This coverage rate is close to the rate reported in Côte d’Ivoire (32.4%) and higher than the Burkina Faso rate (7.9%) and the Togo rate (20.7%) [7]. However, this low coverage observed in sub-Saharan Africa could be explained by several factors. In sub-Saharan Africa, and particularly among people living in rural areas, there has always been, and unfortunately still is, a reluctance to vaccinate in general. Expanded immunization program campaigns are often faced with hesitancy from specific population groups. This uncertainty is mostly linked to various socio-anthropological factors, such as religious beliefs, cultural practices, and defiance attitudes [23,24]. It is in this already difficult context that the COVID-19 vaccination was followed by fake rumors and news that increased doubts. In addition, the vaccines were developed promptly, given the urgency of the situation, compared to the usual timeframes. In addition, for some vaccines, the implementation of new manufacturing technologies, including messenger RNA vaccines, heightened fears. All these factors contributed to increased population hesitation.

It should be noted that the WHO and other partner institutions have played a positive role in making COVID-19 vaccines available through the COVAX initiative for sub-Saharan Africa [25]. COVAX is one of the global initiatives that aims to accelerate the development and manufacture of COVID-19 vaccines and to ensure equitable access to all countries in the world, particularly those with limited resources [26,27].

Our findings also revealed that geographical location and vaccine acceptability were significantly associated. Indeed, the majority of refusals to be vaccinated were recorded in the Cotonou and Abomey-Calavi settings (64.2% and 54.7%, respectively). The probability of COVID-19 vaccine acceptance was two-fold higher for people living in Djougou compared to those living in Cotonou. This result seems surprising, as the outbreak in Benin affected Cotonou the most, and one would have expected a higher acceptability given the experience of more severe cases. This could be explained in part by the high level of doubt about the effectiveness of vaccines in southern Benin. The people living in the Littoral district have access to multiple information sources spreading all kinds of rumors and fake news (misleading information). Previous studies on vaccines have also shown that people often feel that vaccines are not effective [28]. This lack of confidence in vaccines has led to an increased reluctance to receive COVID-19 vaccines, even in developed countries [29]. Distrust of COVID-19 vaccine efficacy may be the result of infodemics, misconceptions, and rumors that immunized people will be infected later [30]. Therefore, there is a need to sensitize and educate the population about vaccines and their developmental stages before administering them to humans.

We also observed that the source of information was a key factor for vaccine acceptance. This factor has also been reported in other studies [18,24,31]. It is noteworthy that the most important sources of information related to COVID-19 among the study population were word-of-mouth information, television, the radio, community and religious leaders, and social media, rather than, for example, government sources and health workers, which is consistent with the literature [32]. Previous studies have shown that people who rely primarily on social media as their main source of information are more likely to be hesitant than those who rely on professional sources of information [33]. Thus, as previously highlighted, social media should be used more effectively as a tool for communicating the right and appropriate information about vaccine strategies, especially against emerging diseases [34,35].

The main reasons for not receiving a COVID-19 vaccine, or not knowing whether to vaccinate, were related to concerns about the safety and side effects of the vaccine. This observation is consistent with other studies conducted since the roll-out of the COVID-19 vaccination program [36,37,38]. The frequently reported fear of possible side effects indicates that risk perception is a major barrier to COVID-19 vaccination uptake. The rapid development of the vaccines also fueled this fear of vaccine safety and reliability. This suggests that the public needs to understand how it has been possible to develop COVID-19 vaccines so quickly while ensuring vaccine safety. This information needs to be communicated clearly to facilitate understanding in a sea of information and misinformation circulating about COVID-19 vaccines [39].

Regarding the history of previous COVID-19 infection, very few participants reported being infected (3.7%). This value should be interpreted with caution, as it was based on the participant’s declaration.

The COVID-19 pandemic is over at the moment of publishing this manuscript. However, understanding the vaccine acceptability factor is relevant for drafting an appropriate vaccine campaign in case the country experiences any other outbreaks of such diseases or when, unfortunately, another pandemic occurs in the future. Our findings suggest that public awareness messages by local authorities are a good predictor of vaccine acceptance by the community.

## 5. Study Strengths and Limitations

To the best of our knowledge, this was the first study to be carried out on this topic in the West African region at the subnational level. The district was included based on its COVID-19 epidemiological data during the pandemic. The sample size and precision are quite enough for the assumption of the representativity and generalizability of our findings. However, due to some limitations, interpretations of our results should be undertaken with caution. The following limitations should be taken into consideration when interpreting our findings. The survey during the school period likely led to the absence of some people during household visits. There were households where the head of the household was not present and the answers were obtained from another member. This would have induced some selection and information biases. Furthermore, vaccination status and vaccine acceptability were both collected based on a declaration. Even if, for the vaccination status, we were able to discriminate between those who declared they were vaccinated and those who were actually vaccinated, this is not the case for vaccine acceptability. Nevertheless, the sensitivity analyses carried out considered the variability and reassured us as to the results obtained.

## 6. Conclusions

We investigated the determinants of the acceptability of the COVID-19 vaccine in Benin. This strengthens the COVID-19 vaccination strategy in Benin. In light of these results, we suggest continuing to inform and raise awareness in the population. The communication strategies during future pandemics or outbreaks should involve local authorities. Additional research using mixed quantitative and qualitative methods is still needed to better understand the COVID-19 vaccine hesitancy. Predictive analytical models are also a promising research portfolio for anticipating similar emerging diseases, particularly outbreak waves, and their impact, and for accelerating vaccination campaigns. Failing to do so may compromise the vaccine progress achieved so far. Further sensitization messages should focus on the safety and efficacy of the vaccines used as part of the COVAX initiative to reassure the population.

## Figures and Tables

**Figure 1 vaccines-11-01104-f001:**
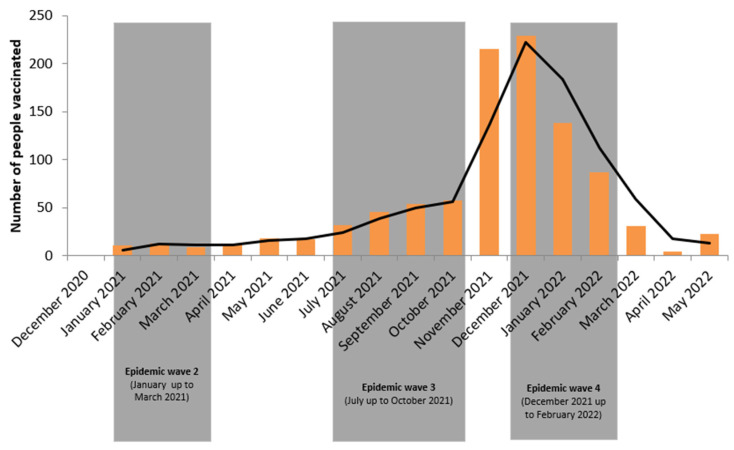
COVID-19 vaccine demand trend.

**Table 1 vaccines-11-01104-t001:** General characteristics of the study population.

Characteristics		Proportion, % (n)
Study districts	Cotonou	20.1% (416)
	Abomey-Calavi	36.6% (758)
	Porto-Novo	29.3% (607)
	Djougou	13.9% (288)
Residence	Urban	59.9% (1241)
	Rural	35.6% (736)
	Semi-rural	4.5% (92)
Age (years)	12–17	2.6% (53)
	18–29	32.7% (677)
	30–39	26.9% (557)
	40–49	16.0% (331)
	50–59	10.8% (224)
	≥60	11.0% (227)
Sex	Female	53.7% (1111)
	Male	46.3% (958)
Number of people living in the household	<4	42.5% (880)
	≥4	57.5% (1189)
Education	None	16.9% (350)
	Primary	26.0% (538)
	Secondary	23.4% (484)
	High secondary	17.1% (354)
	College	16.6% (343)
Ethnic group	Fon and related	30.3% (626)
	Goun	20.3% (421)
	Dendi	6.8% (141)
	Aizo	5.2% (107)
	Others *	30.1% (622)
Marital status	Single	19.6% (405)
	Married	68.3% (1413)
	Divorced	5.0% (104)
	Widowed	7.1% (147)
Religion	Christian	60.4% (1249)
	Muslim	26.9% (556)
	Traditional	7.5% (155)
	Others	5.3% (109)
Channel of COVID-19 information		
Radio	Yes	77.5% (1604)
Television	Yes	69.4% (1435)
Newspaper	Yes	9.4% (194)
Social network	Yes	46.9% (972)
Internet	Yes	18.6% (385)
Healthcare workers	Yes	21.1% (436)
Community political leaders	Yes	31.8% (658)
Religious leaders	Yes	42.6% (882)
Word-of-mouth	Yes	80.1% (1658)
Others	Yes	0.9% (18)

* Others: Adja, Aizo, Batonou, Berba, Mahi, Nago, and Toffin.

**Table 2 vaccines-11-01104-t002:** Medical history of study participants and their knowledge, attitude, and practices related to COVID-19 disease.

Characteristics		Proportion, % (n)
Medical conditions		
High blood pressure	Yes	11.0% (220)
Diabetes	Yes	2.8% (58)
Cardiac disease	Yes	1.0% (21)
Thromboembolic disease	Yes	0.2% (4)
Tumors	Yes	0.3% (6)
Immunodeficiency	Yes	0.3% (6)
Tobacco history	Yes	3.8% (78)
COVID-19 infection	Yes	3.7% (77)
Overall knowledge of COVID-19	Poor	29.6% (612)
	Moderate	30.6% (633)
	Good	39.8% (824)
Knowledge of mode of transmission	Poor	30.3% (627)
	Moderate	37.7% (779)
	Good	32.0% (663)
Knowledge of symptoms	Poor	34.9% (722)
	Moderate	18.8% (389)
	Good	46.3% (958)
Prevention, own behaviors (attitude)	Poor	38.9% (804)
	Moderate	34.5% (713)
	Good	26.7% (552)
Behaviors (practices)	Poor	51.0% (1056)
	Moderate	12.5% (259)
	Good	36.4% (754)
The scale of fear of being infected *	Low (<6)	74.2% (1536)
	Moderate (6–7)	7.3% (151)
	High (≥8)	18.5% (382)

* Scale of fear was notated from 0 to 10, with 0 being the lowest level and 10 the highest level.

**Table 3 vaccines-11-01104-t003:** COVID-19 vaccine status and acceptance of the study participants.

Characteristics		Proportion, % (n)
Declared vaccine status	Not vaccinated	43.2% (895)
	Willing to be vaccinated	8.1% (167)
	Vaccinated	48.7% (1007)
Confirmed vaccine status	No, proof was not accessible	6.8% (141)
	No, proof was not found	17.3% (358)
	No, refused to show proof	0.4% (8)
	Yes, proof was presented	24.2% (500)
	Not applicable *	51.3% (1062)
Vaccine status with proof	No	76.8% (1569)
	Yes	24.2% (500)
Crude vaccine acceptance ^1^	No	32.1% (665)
	Yes	67.9% (1404)
Adjusted vaccine acceptance ^2^	No	56.6% (1172)
	Yes	43.4% (897)

* Not applicable concerns participants who declared that they were not or not yet vaccinated; ^1^ Crude vaccine acceptance is defined by people with a declared vaccination status plus people willing to be vaccinated; ^2^ Adjusted vaccine acceptance is defined by people with a confirmed vaccination status plus people willing to be vaccinated.

**Table 4 vaccines-11-01104-t004:** Factors associated with COVID-19 vaccine acceptance among 2069 study participants in the Beninese population, as determined from a mixed-effect logistic regression analysis.

Factors		Vaccine Acceptance (%), n/N	Crude Analysis		Adjusted Analysis	
			OR (95% CI)	*p*-Value	aOR (95% CI)	*p*-Value
District of residence	Cotonou	61.3% (255/416)	1		1	
	Abomey-Calavi	66.2% (502/758)	1.21 (0.81–1.81)	0.3560	1.23 (0.87–1.73)	0.1205
	Porto-Novo	69.4% (421/607)	1.71 (1.14–2.56)	0.4120	1.59 (1.10–2.29)	0.0041
	Djougou	78.5% (226/288)	2.55 (1.57–4.12)	0.0007	2.70 (1.71–4.28)	0.0002
Residence area	Urban	66.4% (824/1241)	1			
	Semi-rural	77.2% (71/92)	1.51 (0.75–3.08)	0.2410	1.08 (0.86–1.36)	0.2506
	Rural	69.2% (509/736)	1.26 (0.95–1.66)	0.1787	1.91 (0.55–1.5)	0.3156
Gender	Male	70.5% (675/958)	1			
	Female	65.6% (729/1111)	0.79 (0.65–0.96)	0.0189	0.91 (0.97–1.45)	0.2012
Age (years)	12–17	58.5% (31/53)	1			
	18–29	67.1% (454/677)	1.73 (0.95–3.17)	0.4231	1.93 (0.85–2.37)	0.7142
	30–39	67.7% (377/557)	1.85 (1.00–3.40)	0.3450	1.94 (0.75–4.14)	0.6025
	40–49	69.5% (230/331)	1.96 (1.04–3.67)	0.6120	1.69 (0.84–3.71)	0.5014
	50–59	72.8% (163/224)	2.58 (1.33–4.98)	0.8410	1.85 (0.83–4.98)	0.9085
	≥60	65.6% (149/227)	1.67 (0.88–3.19)	0.0747	1.67 (0.98–2.19)	0.0512
Education level	None	63.4% (222/350)	1		1	
	Primary	65.1% (350/538)	1.27 (0.94–1.71)	0.2410	1.21 (0.89–1.66)	0.3541
	Secondary	65.5% (317/484)	1.21 (0.89–1.65)	0.2651	1.13 (0.82–1.57)	0.4120
	High secondary	69.2% (245/354)	1.43 (1.02–2.01)	0.1510	1.26 (0.88–1.81)	0.2520
	College	78.7% (270/343)	2.41 (1.67–3.48)	0.0001	1.88 (1.25–2.81)	0.0305
Religion	Others	65.1% (71/109)	1		1	
	Christian	64.9% (811/1249)	1.06 (0.69–1.64)	0.4521	1.22 (0.42–3.35)	0.7120
	Muslim	75.9% (422/556)	1.52 (0.95–2.46)	0.1241	1.12 (0.82, 1.53)	0.5120
	Traditional	64.5% (100/155)	1.05 (0.61–1.80)	0.0569	1.05 (0.85, 1.82)	0.3121
Scale of fear of being infected *	Low	65.5% (1006/1536)	1		1	
	Moderate	76.2% (115/151)	1.72 (1.15–2.57)	0.0002	1.60 (1.06–2.41)	0.0001
	High	74.1% (283/382)	1.34 (1.03–1.76)	0.0055	1.25 (0.94–1.65)	0.0319
Channel of information						
	Television	70.6% (1013/1435)	1.34 (1.08–1.66)	0.0074	5.43 (1.56–2.3)	0.0081
	Newspaper	74.2% (144/194)	1.35 (0.94–1.94)	0.1060	1.97 (0.61–3.26)	0.1256
	Social network	74.9% (728/972)	1.72 (1.39–2.12)	<0.0001	1.81 (1.25–2.68)	<0.0001
	Internet	77.7% (299/385)	1.77 (1.34–2.34)	0.0001	1.32 (1.15–2.62)	<0.0001
	Healthcare workers	78.9% (344/436)	1.75 (1.33–2.31)	0.0001	1.33 (1.05–2.62)	<0.0001
	Community political leaders	75.4% (496/658)	1.47 (1.16–1.86)	0.0016	1.32 (1.03–1.69)	0.0277
	Religious leaders	73.4% (647/882)	1.38 (1.11–1.72)	0.0043	1.63 (1.05–1.68)	0.0325
	Word-of-mouth	69.7% (1155/1658)	1.30(1.01–1.68)	0.0413	1.47 (1.05–1.88)	0.0452
Medical conditions						
	High blood pressure	71.8% (158/220)	1.29 (0.93–1.77)	0.1225	1.43 (1.03–1.99)	0.0326
	Cardiac disease	42.9% (9/21)	0.34 (0.14–0.84)	0.0199	0.30 (0.12–0.77)	0.0128
	Kidney failure	42.9% (3/7)	0.35 (0.07–1.63)	0.1794	0.25 (0.06–1.43)	0.2135
Overall knowledge of COVID-19	Poor	58.5% (358/612)	1			
	Moderate	66.8% (423/633)	1.52 (1.19–1.94)	0.0051	1.52 (1.19–1.94)	0.0004
	Good	75.6% (623/824)	2.24 (1.76–2.86)	<0.0001	2.24 (1.76–2.86)	<0.0001
Knowledge of the mode of transmission	Poor	64.1% (402/627)	1		1	
	Moderate	70.3% (548/779)	1.46 (1.15–1.87)	0.0042	1.04 (0.80–1.35)	0.0512
	Good	68.5% (454/663)	1.29 (1.01–1.67)	0.0078	0.76 (0.57–1.01)	0.0478
Knowledge of symptoms	Poor	59% (426/722)	1		1	
	Moderate	68.1% (265/389)	1.51 (1.14–1.99)	<0.0001	1.48 (1.11–1.99)	<0.0001
	Good	74.4% (713/958)	1.98 (1.59–2.48)	<0.0001	1.62 (1.24–2.12)	0.0004
Own behaviors (attitude)	Poor	59.7% (480/804)	1		1	
	Moderate	72.2% (515/713)	1.69 (1.35–2.13)	<0.0001	1.41 (1.10–1.79)	<0.0001
	Good	74.1% (409/552)	2.05 (1.58–2.65)	<0.0001	1.69 (1.27–2.24)	0.0006
Behaviors (practices)	Poor	64.8% (684/1056)	1		1	
	Moderate	67.9% (176/259)	1.19 (0.87–1.61)	0.0651	1.39 (0.70–1.77)	0.2317
	Good	72.1% (544/754)	1.49 (1.20–1.87)	0.0017	1.60 (1.28–2.61)	<0.0001

* Scale of fear was notated from 0 to 10, with 0 being the lowest level and 10 the highest level.

## Data Availability

The data presented in this study are available from the corresponding author upon request.

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
