# Peer review of "Assessment of COVID-19 Vaccine Acceptance and Its Associated Factors during the Crisis: A Community-Based Cross-Sectional Study in Benin"

_vaccines, 2023, doi:10.3390/vaccines11061104_

Round 1

Reviewer 1 Report

The authors studied the acceptance of COVID-19 vaccination in Benin. The outcome measure on vaccine acceptance included the willingness to be vaccinated. It would be better to compare between (i) "actual" vaccination and (ii) the willingness to be vaccinated. For example, comparing the factors associated in these two groups.

The COVID-19 pandemic is over at the moment of reviewing this manuscript. Understand the acceptability may be less relevant to the pandemic. The authors are suggested to discuss the messages / findings that may be applicable when, unfortunately, another disease pandemic occurs in the future.

Some more comments regarding the methodology and the presentation of the findings.

lines 57-58. "Vaccination is an effective tool to reduce the burden of COVID-19, but its success depends on the high population vaccine acceptability"
Instead of vaccine acceptability, actual vaccine coverage would be much important. Furthermore, while vaccine coverage affects the public health impacts, vaccine efficacy is also important, particularly in the individual level.

lines 79-80. "The estimated sample size was 2,000 participants to ensure good power." The authors are suggested to elaborate more about the sample size calculation. How did they obtain the estimated sample size? What is the rationale for the estimated sample size of 2,000 participants? What is the "good" power for the achieving the planned sample size.

lines 81-83. "The first stage was a "stratified sampling by district with a proportional inclusion probability" weighted on the density of the population. The second step was "cluster sampling using a general household survey approach"."
May the authors elaborate more about the use of cluster sampling? Cluster sampling means selecting all cases in the selected clusters. What was the unit for the "cluster" in the second step? If each single households are the unit, then it would be simply the stratified sampling with households as the unit.

line 85. "A structured, standardized, and validated questionnaire was administered" How was the questionnaire validated?

lines 85-87. "A structured, standardized, and validated questionnaire was administered face-to-face to one household member randomly selected in the household.Data were collected from both the head of the household and the selected member"
It is a bit confusing. Does it mean that only one randomly selected household member was interviewed and that selected member was asked to provide data for both the head of the household the selected member him/herself?
What if the selected member was the head of the household already?

lines 106-108. "The main outcome was vaccine acceptance defined by participants already vaccinated plus those who did not yet but were willing to be vaccinated"
The authors are suggested to include sensitive analysis for considering "actual vaccination" and "willingness to be vaccinated" separately.

lines 150-153. The number of willingness to be vaccinated was not presented in Table 3. Also, in Table 3, what did "Not yet vaccinated" mean?

Table 4. Please include the p-values for all relevant levels among categorical variables.

The authors may discuss the validity regarding that very few respondents reported being infected (3.7%). Was it consistent with the national statistics when the survey was conducted?

line 183. The study has a sample size of 2069 subjects, rather than a population size.

lines 185-188. I don't quite understand the following statement: "However, it was slightly lower than that reported by Mudenda et al. in Zambia (33.4%)" Either the crude or adjusted vaccine acceptance (67.9% or 43.4%, respectively) was higher than that in Zambia.

The authors may consider providing the questionnaire they used (e.g., as an online supplementary material)

There were some minor grammatical issues spotted.

Author Response

Dear reviewer,

We would like to thank you for providing very valuable comments and suggestions to improve our manuscripts. The manuscript has been reviewed and we were able to provide replies to all of your comments. Please see the point-by-point reply to the reviewer's comments.

Reviewer 2 Report

Manuscript (ID: vaccines-2422785) presents results of investigation of the COVID-19 vaccine acceptance and its associated factors in the Beninese population which could be used to develop accurate communication strategies around Covid-19 immunization.   

But, some issues in this manuscript require major revision:   

  • Lines 42-45: In this sentence, it is mandatory to state that the presented data on the COVID-19 pandemic is conclusive until December 2022.
    • The authors have already mentioned it under reference No. 1 in the list of References.
    • Today, the data on COVID-19 is significantly different, so for the sake of accuracy for readers, it is important to specify the time to which the data refers in that sentence in the text.  
  • Lines 49-50: An appropriate reference should be given for the statement in this sentence. It may also be possible to specify the electronic databases for Benin, for example the relevant Ministry or the national center for public health / epidemiology / center for prevention and control of diseases.
  • Lines 55-56: An appropriate reference should be given for the statement in this sentence.
  • Line 56: Add a new paragraph in which the course of the COVID-19 pandemic in Benin should be briefly described: the total number of inhabitants, the date when the first case of COVID-19 was reported, whether there was a lock-down, when the vaccination against COVID-19 started, which segment of the population is scheduled for vaccination against COVID-19, was the vaccination free, what is the coverage of vaccination against COVID-19, etc.    
  • Line 77: The subsection should be called `Study population / Study sample`.
  • Lines 78-79: As the authors have already stated on Lines 277-279, it would be good to provide such important information in this text in the Methods section, as stated `All participation was voluntary. Written informed consent was obtained from an adult participant over 18 years old in the household. Assent was sought for children from 12 to 17 years.`.      
    • Note - Specify the age in the last sentence, as follows: `Assent was sought for children from 12 to 18 years.`.     
  • Lines 94-102: Table 2 presents data on personal health history. State and explain it in the Methods section. Explain whether such data are self-reported or confirmed by reviewing the appropriate medical documentation.
  • Line 103: List who were the people who collected the data in this study. How many people were there who collected data in this study. State their profession. How long did it take to collect data per 1 respondent. 
  • Lines 84-122: Attach the entire questionnaire as a Supplement file. Also, give an explanation of how and according to which criteria the variables presented in this paper were categorized.    
  • Lines 165-175 and Table 4: Check and correct the following  
    • On Lines 174-175, statistical significance is indicated for `practices`, but on Table 4 significance is reached not for `practices` but for `Own behaviors (attitude)`. Correct this.   
    • Interpret the result for the `Knowledge, mode of transmission` variable.      
  • Lines 176-257: In the Discussion section, the authors presented a comparison of their own results with the results of similar studies in other countries, while providing appropriate explanations for the aforementioned similarities and differences between studies in different populations. A logical flow and comprehensive approach in writing the Discussion is evident.    
  • Line 264: A discussion of the limitations of this study is missing. Add a new paragraph to discuss `Strengths and limitations of the study`.  
  • Line 264: Add a Conclusion section at the end of the paper, in which the most important results of this study should be highlighted.    

The quality of English language is appropriate.  

Author Response

(The authors gave the same response as above.)

Round 2

Reviewer 1 Report

Thanks for addressing my comments.

When preparing the response letter to reviewers' comments, I would suggest the authors to state the changes/revisions explicitly in the letter, in addition to saying that the sentences/sections have been revised/updated (e.g., reply to comment 2). However, the reply to the comments is clear in general.

Reviewer 2 Report

The authors responded correctly to all my comments, gave satisfactory explanations and made all the corrections in the manuscript in an appropriate manner.   

Thanks to the authors.  

The quality of English language is appropriate.